

# Identification of key genes and validation of key gene aquaporin 1 on Wilms' tumor metastasis

Hong Liu[1], Chen Jin[1], Xia Yang[2], Nan Xia[2], Chunzhi Guo[3] and Qian Dong[1]

[1] Department of Pediatric Surgery, Affiliated Hospital of Qingdao University, Qingdao University, Qingdao, Shandong, China
[2] Institute of Digital Medicine and Computer-Assisted Surgery, Affiliated Hospital of Qingdao University, Qingdao, Shandong, China
[3] Department of Thyroid Surgery, Affiliated Qingdao Central Hospital, Qingdao University, Qingdao, Shandong, China

Corresponding author
Qian Dong, 18661801885@163.com

## ABSTRACT

**Background.** Wilms' tumor (WT) is one of the most common solid tumors in children with unsatisfactory prognosis, but few molecular prognostic markers have been discovered for it. Many genes are associated with the occurrence and prognosis of WT. This study aimed to explore the key genes and potential molecular mechanisms through bioinformatics and to verify the effects of aquaporin 1 (AQP1) on WT metastasis.

**Methods.** Differentially expressed genes (DEGs) were generated from WT gene expression data sets from the Gene Expression Omnibus (GEO) database. Gene functional enrichment analysis was carried out with the Database for Annotation, Visualization and Integrated Discovery (DAVID). A protein–protein interaction network (PPI) was constructed and visualized by the Search Tool for the Retrieval of Interacting Genes/Proteins (STRING) database and Cytoscape software. Minimal Common Oncology Data Elements (MCODE) was used to detect the important modules in the PPI network, and the important nodes (genes) in the PPI module were sorted by CytoHubba. RT-qPCR was performed to validate the expression of the key genes in WT. Wound healing and Transwell assays were used to detect the cell migration and invasion abilities of AQP1-overexpressing cells. Phalloidin-iFlour 488 was used to stain the cytoskeleton to observe how AQP1 overexpression affects cytoskeletal microfilament structure.

**Results.** A total of 73 co-expressed DEGs were chosen for further investigation. The importance of homeostasis and transmembrane transport of ions and water were highlighted by functional analysis. Gene regulatory network and PPI network were predicted. MCODE plug identified two important modules. Finally, top five key genes were identified using CytoHubba, including Renin (REN), nephrosis 2 (NPHS2), Solute Carrier Family 12 Member 3 (SLC12A3), Solute Carrier Family 12 Member 1 (SLC12A1) and AQP1. The five key genes were mainly enriched in cell volume and ion homeostasis. RT-qPCR confirmed the expression of the five key genes in WT. AQP1 was validated to be expressed at significantly lower levels in WT than in normal tissue. AQP1 overexpression significantly reduced the migratory and invasive capacity of Wit-49 cells, as evidenced by reducing the scratch healing rate and the number of perforated control cells by Wit-49 cells. AQP1 overexpression also reduced the expression of biomarkers of epithelial-mesenchymal transformation, decreased levels of vimentin and N-cadherin and increased expression of E-cadherin, resulting in decreased formation

of conspicuous lamellipodial protrusions, characteristic of diminished WT cell invasion and migration.

**Conclusion**. Our study reveals the key genes of WT. These key genes may provide novel insight for the mechanism and diagnosis of WT. AQP1 overexpression inhibited invasion, migration, EMT, and cytoskeletal rearrangement of WT cells, indicating that AQP1 plays a role in the pathogenesis of WT.

## INTRODUCTION

Wilms' tumor (WT) is the predominant malignant renal neoplasm in childhood, comprising around 90% of pediatric kidney neoplasms (*Breslow et al., 1993*). Early-stage WT typically presents with subtle or no discernible symptoms, which brings great difficulties to the early diagnosis of the disease. Despite significant advancements in the treatment of WT over the past few decades, there is still a pressing need to improve the survival rate of patients. Although much progress has been made in WT treatment in recent decades, the survival rate for patients with WT still needs to be improved: 2.2% of patients relapse after first-line therapy and up to 25% of survivors report severe late morbidity of treatment (*Suh et al., 2020*; *Termuhlen et al., 2011*). Novel biomarkers, therefore, are urgently needed in order to diagnose the disease early and develop novel target-specific therapies. However, the integration of factors such as genotype, biomarkers, and other risk assessment parameters in WT lags behind compared to other pediatric malignancies. It is very important to improve the prognosis for WT patients by promoting the treatment based on clinical and biological risk factors, and further stratification of current treatment options based on tumor biology (*Treger et al., 2019*; *Zhou et al., 2021*).

In early research on WT, susceptibility loci were identified through genetic linkage studies in familial cases, including 11p13 (WT1), 17q21 (FWT1), 19q13.13 (FWT2), and 11p15 (*Call et al., 1990*; *Karnik et al., 1998*; *Rahman et al., 1996*). With the development of modern genetic technology, *BCOR*, *MAP3K4*, *BRD7*, *CREBBP* and *HDAC4* were involved in histone modifications during kidney formation, and they were verified to have genetic mutations in WT (*Gadd et al., 2017*). Recently, additional putative susceptibility genes, including REST, CHEK2, EP300, PALB2, and ARID1A, have been discovered in WT (*Gadd et al., 2017*; *Mahamdallie et al., 2015*). Nonetheless, the precise biological mechanisms responsible for the roles of these genes in WT are still not fully elucidated. Clear molecular mechanisms are yet to be established. Investigating the key genes associated with WT development will contribute to a deeper understanding of biological mechanisms, early diagnosis, and prognostic implications for WT.

With the progress of molecular biology and computer science and technology, bioinformatics has developed rapidly, leading to the emergence of genomic concepts such as those revealed by genomic, transcriptomic, proteomic, and metabolic analyses. GEO (Gene Expression Omnibus) serves as a comprehensive oncology database, encompassing vast
sequencing and clinical data pertaining to diverse tumor types. In contemporary research, the analysis of data matrices using bioinformatics has become a prevalent approach for identifying genes with differentially expressed genes (DEGs) and performing diverse analytical investigations. The integration of multiple databases enables the amalgamation of data from independent studies, thereby augmenting the sample size for analysis and enhancing the robustness and precision of the findings. Based on the aforementioned considerations, we collated patient data afflicted with Wilms tumor (WT) from the GEO database to facilitate further analysis using bioinformatics tools and experimental validations.

In this study, bioinformatics analysis was used to explore the key genes and the underlying molecular mechanisms of WT. Also, we wanted to conduct preliminary validation of the expression levels and functional mechanisms of key genes in WT.

## MATERIALS AND METHODS

### Materials

The gene expression profile data sets with accession numbers GSE57269, GSE73209 and GSE66405 were found from the Gene Expression Omnibus (GEO) database (http://www.ncbi.nlm.nih.gov/geo/) (*Edgar, Domrachev & Lash, 2002*). Previously, *Shukrun et al. (2014)* submitted chip data (GEO accession GSE57269) to Platform GPL15207, *Karlsson et al. (2016)* submitted chip data (GEO accession GSE73209) to Platform GPL10558, and *Ludwig et al. (2016)* submitted chip data (GEO accession GSE66405) to Platform GPL17077. The GSE57269 data set contains 7 Wilms' tumor samples (Registration number: GSM1378119~GSM1378125) and 1 normal kidney sample (Registration number: GSM1378118). The GSE73209 data set contains 32 Wilms' tumor samples (Registration number: GSM1888569~GSM1888600) and 6 normal kidney samples (Registration number: GSM1888603~GSM1888608). The GSE66405 data set contains 28 Wilms' tumor samples (Registration number: GSM1621548~GSM1621575) and 4 normal kidney samples (Registration number: GSM1621576~GSM1621579).

### Methods

#### DEG identification

To identify differentially expressed genes (DEGs) in Wilms' tumor (WT), the online tool GEO2R (http://www.ncbi.nlm.nih.gov/geo/geo2r/) from the Gene Expression Omnibus (GEO) database was utilized. GEO2R enables the comparison of multiple sample groups within a GEO series (*Barrett et al., 2013*). In this study, the DEGs were screened using GEO2R from three WT sample gene chip datasets: GSE57269, GSE73209, and GSE66405. To reduce false positives, the $p$ values were adjusted using the Benjamini–Hochberg method to control the false discovery rate (FDR). The selection criteria for DEGs included a threshold of |log2fold-change (FC)| $\geq 1.5$ and adjusted $p$ value $< 0.05$. DEGs between WT samples and normal samples were identified by comprehensive analysis. The online tools were used to draw the volcano map of DEGs in two data sets. Venn diagrams (*Pirooznia, Nagarajan & Deng, 2007*) of co-expressed DEGs were drawn by online tools.

## GO and KEGG pathway enrichment analysis

Enrichment analysis is a commonly used method to analyze gene sets derived from genomic studies and identify their associated biological functions and signaling pathways. For the analysis of gene function and pathway enrichment, we utilized the Database for Annotation, Visualization, and Integrated Discovery (DAVID), (https://david.ncifcrf.gov), a comprehensive web-based resource (*Huang da, Sherman & Lempicki, 2009*). This resource facilitated Gene Ontology (GO) and Kyoto Encyclopedia of Genes and Genomes (KEGG) analyses in our study. Significance was determined based on an adjusted $p$ value threshold of < 0.05.

## PPI network and key genes were selected

The protein–protein interaction network (PPI) was investigated using the Search Tool for the Retrieval of Interacting Genes/Proteins (STRING) database (https://string-db.org/) (*Szklarczyk et al., 2019*). The DEGs were subjected to analysis in STRING, leading to the construction of the PPI network. To ensure reliability, a minimum interaction score of 0.7, denoting high confidence, was employed as the cutoff threshold. In the PPI network, high confidence score was used as indicators to filter and determine the reliability and repeatability of protein interactions. Setting 0.7 as the minimum value was done to ensure that the resulting PPI network has a higher level of confidence and biological significance. The PPI network was visualized using Cytoscape v3.7.2 software (https://cytoscape.org/) (*Shannon et al., 2003*). To identify significant modules within the network, we utilized the Molecular Complex Detection (MCODE) plugin, a tool integrated with Cytoscape. The module identification process involved applying specific criteria, including a degree cutoff of 2, a node score cutoff of 0.2, a K-core value of 2, and a maximum depth of 100 (*Bader & Hogue, 2003*). By applying the aforementioned criteria, it is possible to filter and identify modules that exhibit high connectivity, dense interactions, and biological relevance, enabling a better understanding of protein interactions and biological functions. We employed the CytoHubba plugin in Cytoscape (*Chin et al., 2014*) to rank and explore crucial genes within the PPI network modules. For the top 5 key genes, we conducted enrichment analyses using DAVID, applying a significance criterion of an adjusted $p$ value < 0.05, for both GO and KEGG enrichment analyses.

## Gene regulatory network analysis

We utilized the eXpression2Kinases (X2K) web-based application (*Clarke et al., 2018*) (https://amp.pharm.mssm.edu/X2K/) to examine the participation of upstream cellular signaling pathways and establish the regulatory connections among transcription factors (TFs), protein kinases, and their target genes. By employing default parameters, X2K was utilized to examine the upstream regulatory network governing the gene expression regulation of the upregulated genes. The inferred network, which consisted of co-expressed differentially expressed genes, was constructed and visualized.

## Cell culture and tissues

WT Wit-49 cells and human renal tubular epithelial cells (HK-2) were obtained from the Cell Bank of the Chinese Academy of Sciences. These cells were cultured in a medium

supplemented with 10% fetal bovine serum (FBS) under standard conditions: 37 °C and 5% CO2. From May 2013 to May 2021, WT and matched normal renal tissues from 32 children with pathologically confirmed WT were collected from the Affiliated Hospital of Qingdao University. None of these patients had received adjuvant radiotherapy or chemotherapy prior to surgical intervention. This study was conducted in accordance with the ethical guidelines set by the Ethics Committee of the Affiliated Hospital of Qingdao University (Registration No. QYFY WZLL 27451). Prior written informed consent was obtained from the parents or legal guardians of all the participating children.

### RNA extraction and quantitative real-time PCR (qRT–PCR) analysis

Total RNA from the gastrocnemius was extracted using TRIzol reagent (Invitrogen, Grand Island, NY, USA), and reverse transcription of RNA into complementary DNA (cDNA) was performed using a reverse transcription kit (TaKaRa, Shiga, Japan) and oligo (dT) primers. qRT–PCR (*Ying et al., 2017*) was conducted using gene-specific primers and SYBR Green qPCR mix (Thermo Fisher, Waltham, MA, USA). The primer pairs were as follows: *Renin* forward: 5′-CTCTACACTGCCTGTGTGTATC-3′, *Renin* reverse: 5′-CACTGACTGTCCCTGTTGAATA-3′. *NPHS2* forward: 5′-CTGTGAGTGGCTTC-TTGTCCTC-3′, *NPHS2* reverse: 5′-CCTTTG GCTCTTCCAGGAAGCA-3ʹ. *SLC12A3* forward: 5′-TGGACGACCATTTC CTACCTGG-3′, *SLC12A3* reverse: 5′-CACTCGGTGAAG TTCCAGCCAT-3′. *SLC12A1* forward: 5′-AGGCTCTTTCCTACGTGAGTGC-3′, *SLC12A1* reverse: 5ʹ-GCCACTGTTCTTGGTAAAGGCG-3′. *AQP1* forward: 5′-CTGTGGGATTAA CCCTGCTCG-3′, *AQP1* reverse: 5′-GAAG -CTCCTGGAGTTGATGTCG-3′. GAPDH was used as a reference for each sample.

### Western blotting

Western blotting (WB) was conducted following a previously described protocol (*Barik et al., 2014*). Total protein was extracted using RIPA Lysis Buffer (Beyotime, China), and the protein concentration was determined using the Bicinchoninic Acid (BCA) Protein Assay (Beyotime, China). SDS-PAGE was performed to separate the protein samples, which were then transferred to a nitrocellulose membrane, blocked, and probed with primary antibodies, including anti-AQP1, anti-Vimentin (1:1000; Abcam, Cambridge, UK), E-Cadherin (1:1000; CST, USA), N-Cadherin (1:1000; CST, USA), and anti-β-actin (1:1000; Abcam, Cambridge, UK). The membranes were subsequently incubated with horseradish peroxidase (HRP)-conjugated secondary antibodies (1:1000; Invitrogen, Waltham, MA, USA). Immunoreactive protein bands were visualized using an HRP chemiluminescence detection reagent (ECL; Thermo Fisher Scientific), and the blots were imaged using a ChemiDoc MP imaging system (Bio-Rad, Hercules, CA, USA). The grayscale values were quantified by ImageJ.

### Immunohistochemical (IHC) staining

Paraffin sections of WT and matched normal kidney tissues (more than five cm from the edge of the cancer focus) were stained with an anti-AQP1 antibody. Before staining, the paraffin-embedded tissue sample was cut into 4 μm sections, dewaxed, rinsed in anhydrous xylene, and rehydrated by gradient ethanol for antigen repair (*Al-Dhohorah et al., 2016*).

To block endogenous peroxidase activity, the sliced samples were exposed to 3% hydrogen peroxide, followed by three washes with phosphate-buffered saline (PBS). Subsequently, the samples were sealed with a 5% FBS-PBS solution for 15 min and then incubated overnight at 4 °C with the primary antibody. On the following day, the samples were incubated with the secondary antibody for 60 min. Finally, 3,3′-diaminobenzidine (DAB) was detected by immunohistochemical Avidin-Biotin Complex (ABC) staining. Under a light microscope, the cytoplasm or nucleus was stained yellow or brown for positive staining, and samples without staining were negative. The images were taken at 100× and 200× magnification. ImageJ software was used to analyze the optical density of the immunohistochemical photos. Three pathological sections of each specimen were taken to detect the cumulative integrated optical density (IOD), and the average value was calculated.

### Transfection

Wit-49 cells were maintained in a 37 °C incubator with 5% $CO_2$ until they reached approximately 70% confluence. The AQP1 cDNA (NM_198098) plasmid or a negative control plasmid was obtained from GeneChem (Shanghai, China) and transfected into Wit-49 cells using Lipofectamine 3000 (Invitrogen, CA, USA). After 24 h of incubation, the medium was replaced, and the corresponding functional assays were performed 24 h later. Protein extraction was conducted 48 h after transfection, and the overexpression of AQP1 was confirmed by Western blotting.

### Wound healing to detect cell migration

Confluent monolayers of *AQP1*-overexpressing cells and negative control cells were subjected to a defined wound by manually removing a 300–500 µm strip of cells using a standard 200 µl pipette tip. The wounded monolayers were washed twice to eliminate nonadherent cells. The speed of wound healing was quantified by measuring the average linear migration rate of the wound edges over a 24-hour period. Cell migration was observed under a light microscope, and cell mobility was calculated using ImageJ analysis software as follows: Cell mobility = (0 h scratch area-24 h scratch area)/0 h scratch area. The average was calculated from 3 independent experiments.

### Transwell assays to detect cell invasion

For the analysis of invasion ability, chambers coated with Matrigel (Corning, NY, USA, 354480) were utilized to culture *AQP1*-overexpressing cells and negative control cells. After incubating for 24 h, the cells that migrated through the Transwell chambers were fixed using methanol and stained with Giemsa. The number of migrated cells was quantified by examining five randomly chosen fields per well under a microscope. The average value was calculated based on data obtained from three independent experiments.

### Phalloidin-iFlour 488 staining of the cytoskeleton

Cells were seeded onto cell climbing tablets and cultured for 24 h. After that, they were fixed using 4% paraformaldehyde for 20 min. To enhance cell permeability, a 0.1% Triton X-100 solution was applied for 15 min. For fluorescence staining of F-actin, coverslips were treated with Phalloidin-iFlour 488 reagent (ab176753) (1:1000; Abcam, Cambridge,

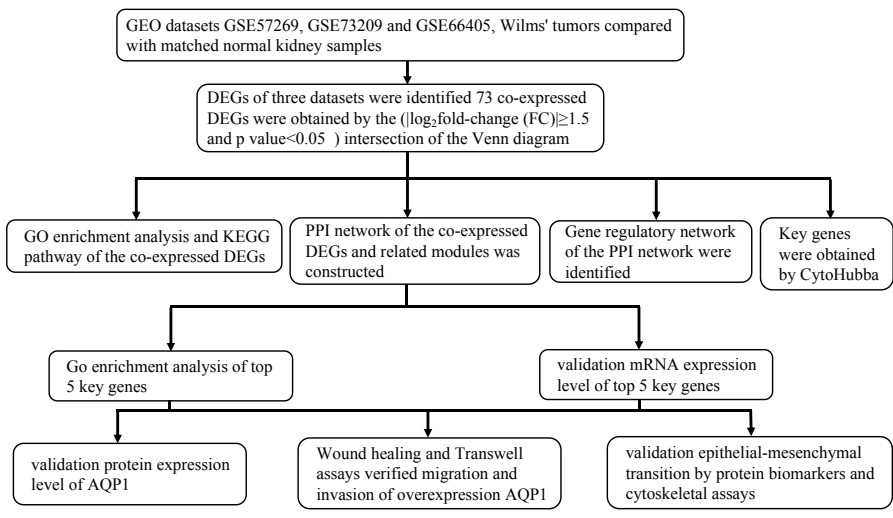

**Figure 1**    **Research design flowchart.**

UK) at room temperature for 45 min, while being protected from light. After washing the cells three times with PBS, nuclear staining was performed using DAPI in the fluorescence assay. Finally, all cells were observed using a Laser Scanning Confocal Microscope (LSCM) (Nikon, Tokyo, Japan) following mounting. Phalloidin-iFlour 488 staining of F-actin revealed the cytoskeleton, which appeared as green fluorescence, and DAPI staining was used to stain the nuclei. Then, we overlapped the two parts.

## Statistical analysis

The mean and standard deviation (SD) were used to present the data. Statistical analysis was performed using SPSS version 22.0 software (SPSS, Chicago, IL, USA). T tests were conducted for statistical evaluation, and a $p$ value < 0.05 was considered indicative of significant differences.

## RESULTS

### Identification of WT-related genes

The research flowchart of this study is shown in Fig. 1. GSE57269 contains 7 WT samples and 1 normal kidney sample, GSE73209 contains 32 WT samples and 6 normal kidney samples, and GSE66405 contains 28 WT samples and 4 normal kidney samples. A total of 5128 DEGs in WT were screened, including 1284 upregulated genes and 3844 downregulated genes (|log2fold-change (FC)| ≥1.5 and adjusted $p$ value < 0.05) (Figs. 2A, 2B, 2C). The intersection of the three groups of DEGs contained 73 genes (co-expressed DEGs) (Fig. 2D).

### GO and KEGG pathway enrichment analysis of the co-expressed DEGs

In order to gain a deeper understanding of the co-expressed DEGs, we conducted enrichment analyses for GO and KEGG pathways. The GO enrichment analysis included

Peer J

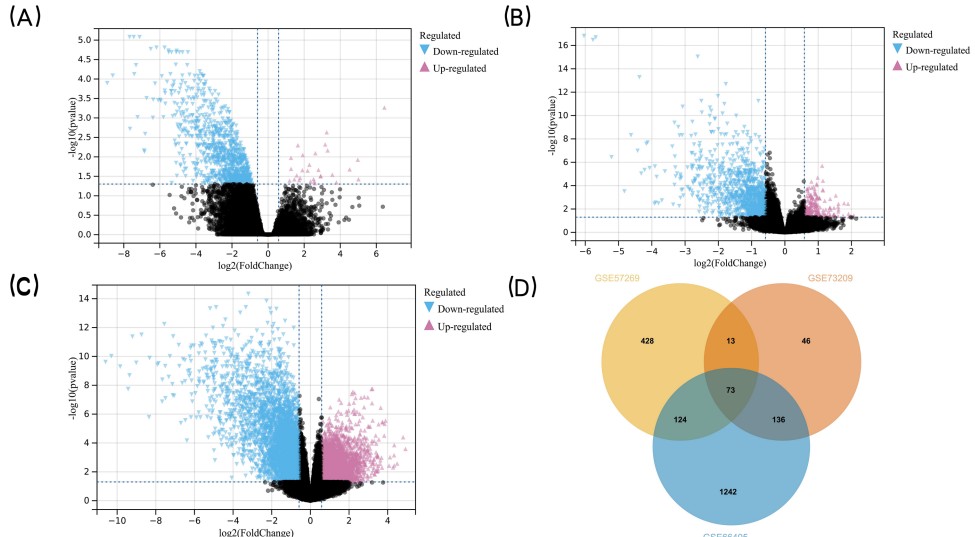

**Figure 2** **Detection of DEGs in WT.** (A) GSE57269 and (B) GSE73209 and (C) GSE66405 (D) their intersection. (A–C) The volcano plots show the difference in gene expression in Wilms' tumor tissues *versus* normal kidney tissues. The x axis indicates the log2 fold changes in gene-expression level between groups, with larger positive values representing genes with higher expression in Wilms' tumors (A: 245 genes, B: 33 genes, (C) 1,006 genes, shown in reddish purple points) and larger negative values representing genes with lower expression in Wilms' tumors (A: 1,810 genes, B: 332 genes, C: 1,702 genes, shown in blue points). The y axis shows the −log10 of the p value for each gene, with larger values indicating greater statistical significance. The two vertical lines are the 1.5-fold change boundaries and the horizontal line is the statistical significance boundary ($p < 0.05$). Black points represent genes from Wilms' tumor tissues and normal kidney tissues for which there was no significant difference in gene expression. (D) Venn diagram was used to determine the intersection between the three datasets to obtain DEGs in common. The three datasets intersected the statistically significant DEGs screened to obtain a total of 73 co-expressed DEGs. The DEGs in two intersecting datasets were 13 genes, 124 genes, and 136 genes respectively.

categories such as biological process (BP), cell component (CC), and molecular function (MF). The results revealed that the co-expressed DEGs were primarily associated with BP terms such as "sodium ion homeostasis", "sodium ion transmembrane transport", and "multicellular organismal water homeostasis." In terms of CC, they were enriched in terms such as "extracellular exosome" and "apical plasma membrane". As for MF, the DEGs were mainly involved in "protein binding" and "sodium: potassium: chloride symporter activity" (Fig. 3A). KEGG pathway analysis identified significant enrichment of the co-expressed DEGs in processes such as "Metabolic pathways", "Proximal tubule bicarbonate reclamation", and "Aldosterone-regulated sodium reabsorption" (Fig. 3B). Taken together, these findings highlight that the co-expressed DEGs may be associated with homeostatic and transmembrane transport of intracellular water and ions, mainly involved in metabolism-related signaling pathway.

## Key genes were selected by the PPI network

The STRING database was applied to construct a protein-protein interaction (PPI) network comprising 73 co-expressed DEGs to explore core modules and key genes, both of which could play a central role in the network. The PPI network, including 48 nodes (genes) and
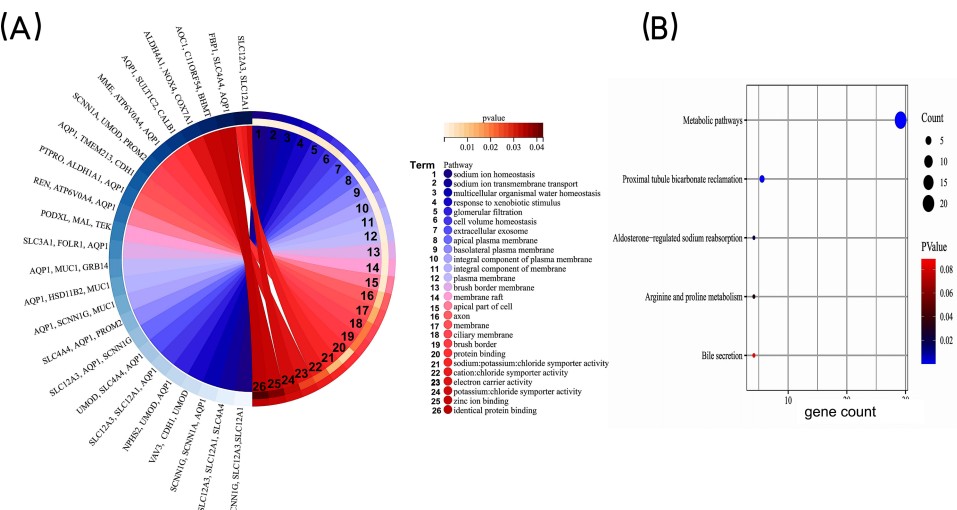

**Figure 3   GO enrichment analysis of co-expressed DEGs.** (A) GO enrichment analysis of the co-expressed DEGs. GO, Gene Ontology. (B) KEGG pathway analysis of co-expressed DEGs. The bubble size is the number of differentially enriched genes, and the bubble color represents the *p* value of the enrichment significance. KEGG, Kyoto Encyclopedia of Genes and Genomes.

93 edges (interactions), was further analyzed by two built-in algorithms within Cytoscape (Fig. 4A). MCODE identified highly interconnected protein subgraphs from PPI network, which often represent functionally relevant protein complexes. 2 modules were retrieved from the PPI network constructed using co-expressed DEGs (Figs. 4B–4C). Among them, the module 1 included seven nodes (genes) and 11 edges (interactions), and the clustering score (density multiplied by the number of members) was 3.667; the module 2 included five nodes (genes) and seven edges (interactions), and the clustering score (density multiplied by the number of members) was 3.5. To rank the important genes within the co-expressed DEGs, the CytoHubba plug-in in Cytoscape was employed based on the degree centrality (Fig. 4E). The top five key genes, identified based on their node degree, were *REN, NPHS2, SLC12A3, SLC12A1,* and *AQP1*.

## X2K construction and visualization of the upstream regulatory network

We used X2K to construct and visualize the upstream regulatory network of the co-expressed DEGs (Fig. 4D). We computationally predicted the involvement of upstream cell signaling pathways and identified the regulatory associations between TFs, protein kinases, and their target genes. E2F Transcription Factor 1 (E2F1) and CAMP Responsive Element Modulator (CREM) were the most important TFs that regulated the co-expressed DEGs. In addition, Mitogen-Activated Protein Kinase 14 (MAPK14) and cyclin-dependent kinase 4 (CDK4) were identified as the most important kinases controlling the co-expressed DEGs. Together, the prediction of the upstream regulatory network controlling the co-expressed DEGs could help us to conduct in-depth mechanism research.

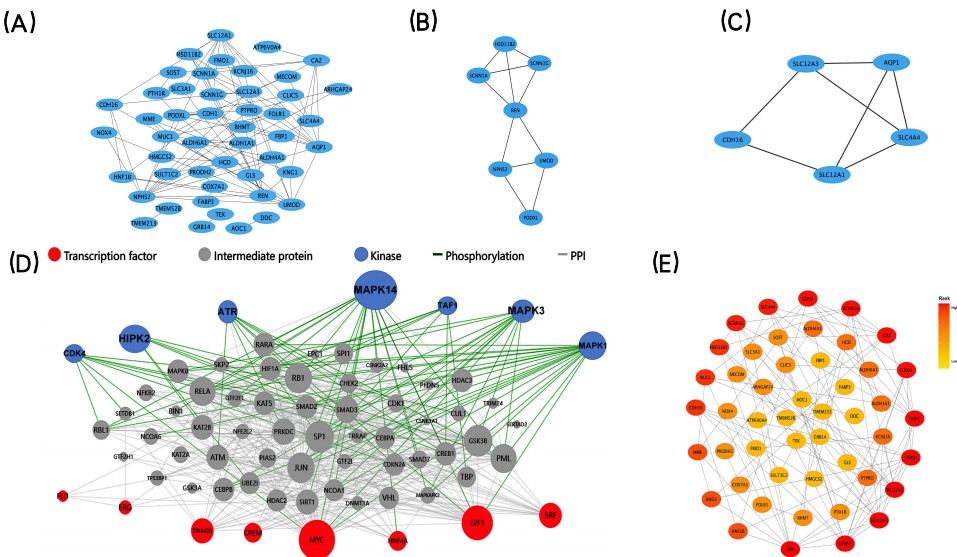

**Figure 4 PPI networks, two significant modules of the PPI network, Upstream regulatory network of the co-expressed DEGs and CytoHubba find the key genes.** (A) A graphic representation of the PPI networks based on co-expressed DEGs. (B–C) Graphic representation of two significant modules of the PPI network. (C) Module 1, D: Module 2. (D) Upstream regulatory network of the co-expressed DEGs. Blue points represent enrichment for kinase likely to regulate co-expressed DEGs. Red points represent enrichment for transcription factors likely to regulate co-expressed DEGs. Gray points represent co-expressed DEGs. (E) A graphic representation of the co-expressed DEGs network based on degree. The CytoHubba in Cytoscape to find the key genes. Circle color represents gene rank.

## GO enrichment analysis of the top five key genes

The functional annotations of top five key genes were performed by DAVID, GO analysis results revealed of BP, CC and MF with *p* value (Table 1).

## Validation of top five key genes expression

In the GEO dataset analysis, *REN, NPHS2, SLC12A3, SLC12A1, AQP1* were significantly downregulated in WT tissues. To validate this finding, the expression levels of the top 5 key genes in WT Wit-49 cells and HK12 normal kidney cells, we performed RT-qPCR. The results revealed that the mRNA expression levels of top five key genes were all downregulated in the WT Wit-49 cells. No statistically significant relationship existed of *REN* and *SLC12A1* (Fig. 5A).

To further investigate the expression levels of AQP1 in renal tubules and WT tissues. Western blotting showed that the expression of AQP1 protein in the WT Wit-49 cells was significantly lower than that in normal renal tubular epithelial HK2 cells (Fig. 5B). We used immunohistochemical (IHC) staining to detect the expression levels in normal renal tissues and in WT tissues. AQP1 was expressed in different parts of the tissues, showing a brown color, and was located in the cytoplasm and cell membrane. AQP1 was expressed in glomeruli, renal tubules, and cancerous tissues and was clearly stained in normal renal tissues. The AQP1 level in pathological sections was quantified by ImageJ software, and the average optical density (AOD) was obtained. The results showed that the normal renal

**Table 1  GO analyses of the key genes.**

| Term | P value | Genes |
|---|---|---|
| **Biological Process** | | |
| cell volume homeostasis | 6.72E−06 | SLC12A3, SLC12A1, AQP1 |
| cellular response to inorganic substance | 8.26E−04 | SLC12A3, AQP1 |
| chloride ion homeostasis | 0.00227132 | SLC12A3, SLC12A1 |
| sodium ion homeostasis | 0.00350859 | SLC12A3, SLC12A1 |
| glomerular filtration | 0.00392075 | NPHS2, AQP1 |
| **Cellular Component** | | |
| apical plasma membrane | 0.00196873 | SLC12A3, SLC12A1, AQP1 |
| plasma membrane | 0.00450275 | SLC12A3, NPHS2, SLC12A1, REN, AQP1 |
| extracellular exosome | 0.00459903 | SLC12A3, NPHS2, SLC12A1, AQP1 |
| apical part of cell | 0.01410947 | REN, AQP1 |
| membrane | 0.02068773 | SLC12A3, SLC12A1, REN, AQP1 |
| **Molecular Function** | | |
| sodium: potassium: chloride symporter activity | 6.34E−04 | SLC12A3, SLC12A1 |
| cation: chloride symporter activity | 0.00126847 | SLC12A3, SLC12A1 |
| potassium: chloride symporter activity | 0.00190225 | SLC12A3, SLC12A1 |

tissues had a higher AOD value than the WT tissues. The protein levels of AQP1 in WT tissues, which were detected by IHC, also confirmed that the protein-level expression of AQP1 is lower in WT (Fig. 5C).

## Overexpression of *AQP1* suppresses the migration and invasion of WT cells *in vitro*

The functional evaluation of *AQP1* in WT cells was performed using an overexpression plasmid, which resulted in a substantial increase in *AQP1* expression in the *AQP1* overexpression group (Fig. 6A). Wound healing and Transwell assays were conducted to investigate the impact of *AQP1* on WT cell migration and invasiveness. The findings revealed that *AQP1* overexpression significantly reduced the migratory and invasive capacity of Wit-49 cells (Fig. 6B). Notably, the scratch healing rate in the *AQP1* overexpression group $(32.90 \pm 5.00)$% was significantly lower than that in the negative control group $(78.70 \pm 4.70)$% $(t = 6.709, P < 0.01)$, indicating that AQP1 overexpression significantly inhibited the migration ability of WT cells. The results of Transwell assays showed that the number of perforated cells was significantly lower in the *AQP1* overexpression group $(179.00 \pm 37.74)$ than in the negative control group $(487.67 \pm 41.72)$ $(t = 5.486, P < 0.01)$, suggesting that *AQP1* overexpression significantly inhibited the invasion capacity of WT cells. To investigate the potential impact of *AQP1* overexpression on tumor metastasis, the influence of *AQP1* on epithelial-mesenchymal transition (EMT) was assessed using Western blotting. The results demonstrated that the *AQP1* overexpression group exhibited decreased levels of Vimentin and N-Cadherin, while the expression of E-Cadherin showed an opposite trend compared to the negative control group (Fig. 6A). Collectively, these findings suggest that *AQP1* overexpression suppresses the migration, invasion, and EMT of WT cells.

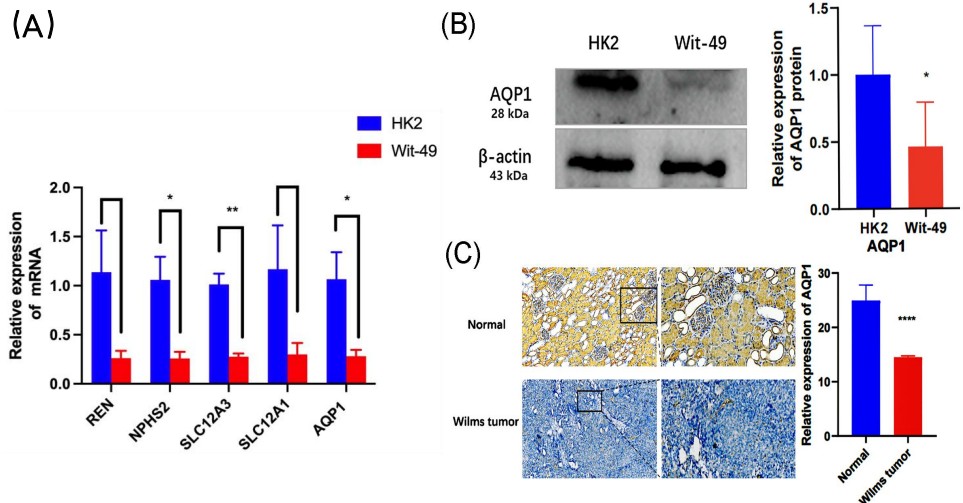

**Figure 5 The mRNA expression levels of the top five key genes and the expression of AQP1 in protein level.** (A) Comparison of the levels of REN, NPHS2, SLC12A3, SLC12A1, AQP1mRNA between normal renal tubular epithelial HK2 cells and WT Wit-49 cells ($n = 3$). The mRNA expression levels of REN, NPHS2, SLC12A3, SLC12A1, AQP1 were all downregulated in the WT Wit-49 cells, but there was not a statistically-significant difference of REN and SLC12A1. (B) Semiquantitative immunoblots of AQP1 from normal renal tubular epithelial HK2 cells and WT Wit-49 cells. Immunoblots were reacted with anti-AQP1 and revealed a 28-kDa band representing AQP1 (upper panel) and a 43 kDa band representing β-actin (lower panel). Quantitative analysis of the relative expression level of AQP1 in normal renal tubular epithelial HK2 cells and WT Wit-49 cells. ($n = 3$). (C) The expression of AQP1 in WT and normal renal tissues (magnification 100). Quantification of the IHC intensity in WT and normal renal tissues. The intensity of the IHC images was measured and quantified by ImageJ. Statistical analysis was conducted using an unpaired $t$ test. $^*p < 0.05$, $^{**}p < 0.01$, $^{****}p < 0.0001$.

The expression of F-actin in the negative control group increased significantly and metastasized to the pericellular periphery. Moreover, the cell volume increased, and lamellipodia, filopodia and stress fibers appeared. Together, F-actin expression was lower, and lamellipodia and filopodia formation decreased in the *AQP1* overexpression group. By regulating the remodeling and dynamic changes of the cytoskeleton, cells were able to alter their shape, generate crawling pseudopods, and reorganize and rearrange microfilaments and microtubules, which could guide directional cell migration. The cell volume increased, and lamellipodia, filopodia, and stress fibers appeared, these changes collectively indicate an increase in cell migration capability. Overall, *AQP1* overexpression decreased the formation of lamellipodia and filopodia, thereby inhibiting cell migration (Fig. 6C).

## DISCUSSION

WT is a prevalent solid tumor in pediatric patients, characterized by challenging early diagnosis. An urgent need exists to improve risk assessment and find a dependable biomarker for precise evaluation of WT patients. In this study, bioinformatics analysis of the gene chip data sets GSE57269, GSE73209 and GSE66405 was first performed by GEO2R. We identified 73 co-expressed DEGs. CytoHubba used PPI network to perform

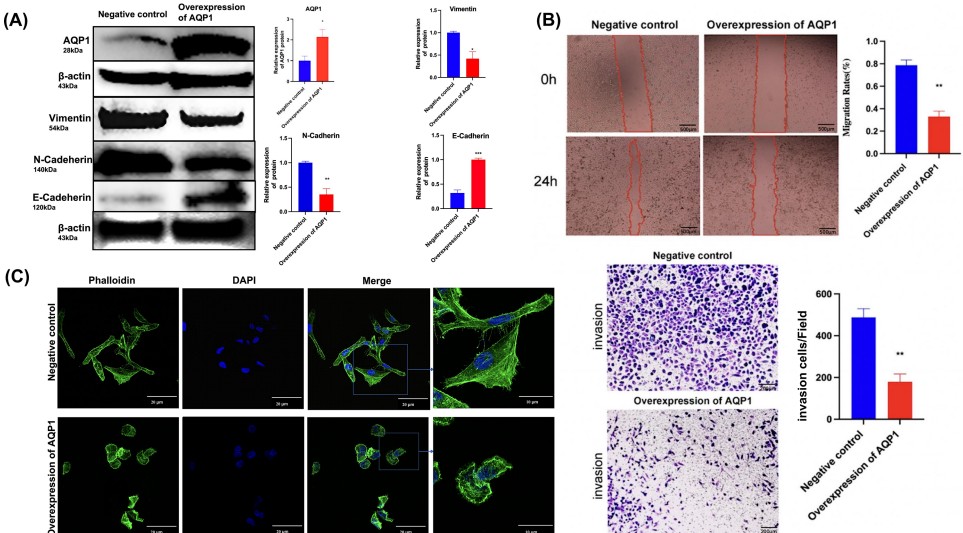

**Figure 6 Overexpression of *AQP1* suppresses the migration and invasion of WT cells *in vitro*.** (A) The overexpression plasmid targeting *AQP1* was used, and the upregulation efficiency was measured by Western blotting. Overexpression of *AQP1* inhibits epithelial–mesenchymal transition, as shown by the expression of Vimentin. (B) Wound healing assays and Transwell assays indicated that *AQP1* significantly decreased the migration and invasive capacity of WT cells. Wound healing assays scale bar 500 $\mu$m, Transwell assays scale bar 200 $\mu$m. (C) Cytoskeletal assays showed that more conspicuous lamellipodial protrusions were formed in the negative control group than in the *AQP1* overexpression group. Objective lens, magnification, '60. Cell nuclei are stained with DAPI. All experiments were repeated three times. Statistical analysis was conducted using an unpaired $t$ test. $^*p < 0.05$, $^{**}p < 0.01$.

multidimensional analysis on co-expressed DEGs. We ranked co-expressed DEGs based on Degree and identified the top 5 key genes: *REN, NPHS2, SLC12A3, SLC12A1,* and *AQP1*.

It has been well documented that *REN* is expressed and plays important roles in WT. *McKenzie et al. (1996)* confirmed that WT consisted of a mixture of cells expressing the renin gene, including both tumor cells and perivascular stromal cells. *Steege et al. (2008)* demonstrated that the Wilms' tumor protein WT1 participates in renin gene regulation. Mutations in the *NPHS2* can disrupt the normal formation and operation of the kidney's filtration system, leading to the onset of proteinuria and related symptomatic manifestations. The Wilms' tumor suppressor 1 (WT1) mutations are thought to significantly contribute to the onset of Wilms' tumor (*Huff, 1998*). The conditional deletion of WT1 specifically in podocytes (WT1fl/fl;Nphs2-Cre) resulted in WT1 expression being restricted to the capillary loop stage, while no presence was observed in mature glomeruli. The kidneys demonstrated a notable decline in mature glomeruli and proximal convoluted tubule count, indicating the critical role of WT1 in regulating podocyte maturation and maintaining proper renal filtration function. Furthermore, WT1 has been found to regulate the expression of Nphs2, a conserved target gene in podocytes (*Dong et al., 2015*). The proteins encoded by the genes *SLC12A3, SLC12A1* and *AQP1* are all transmembrane transport proteins. Their main function is to facilitate the transportation of various substrates across biological membranes. *SLC12A3* and *SLC12A1* are two different members

of the solute carrier family 12 gene family. Experimental investigations into the renal salt transport molecules *SLC12A3* and *SLC12A1* have revealed that a significant portion of the identified mutations lead to impaired transport functionality. These mutations disrupt transport by affecting various aspects, including biosynthetic processing, trafficking, ion transport, and regulation. *SLC12A1* mutation can lead to Bartter syndrome, while *SLC12A3* mutation primarily result in Gitelman syndrome (*Welling, 2014*).

AQP1, the first aquaporin identified and extensively studied in the AQP family (*Denker et al., 1988*), has been detected in various types of human malignancies (*Chen et al., 2006*; *Saadoun et al., 2002*; *Vacca et al., 2001*). In Renal Cell Carcinoma (RCC), *AQP1* expression is generally low (*Ticozzi-Valerio et al., 2007*), and its association with overall survival (OS) has been well-established. Among the key genes identified through bioinformatics analysis, *AQP1* has been validated as an important biomarker for RCC (*Morrissey et al., 2015*). Furthermore, *AQP1* has been identified as an autonomous prognostic factor and a prognostic-related factor in clear cell renal cell carcinoma (ccRCC) (*Li et al., 2022*; *Wang et al., 2022*). Yet, the expression and functional implications of *AQP1* in WT are not well understood. Our research further verified the expression of *AQP1* in WT and its effect on cell migration in WT cells. We confirmed *AQP1* expression at both the mRNA and protein levels. Our findings indicated that elevated levels of *AQP1* exert inhibitory effects on the migratory and invasive capacities of Wilms' tumor cells, and the above effects may be related to affecting the cytoskeletal microfilament structure.

The involvement of *AQP1* in cancer migration and invasion is of great significance, as these processes play pivotal roles in human carcinogenesis. It is noteworthy that *AQP1* has demonstrated divergent effects on migration in different types of cancer (*Moon, Moon & Kang, 2022*). Studies have indicated that *AQP1* overexpression in B16F10 melanoma cells and 4T1 mammary gland tumor cells promotes cell migration and lamellipodia formation by enhancing osmotic water permeability across the cell membrane (*Hu & Verkman, 2006*; *McLean et al., 2005*). Furthermore, Jiang et al. conducted a study to examine the expression of *AQP1* in diverse cancer cell lines, including an HT20 colon cancer cell model, revealing that the introduction of *AQP1* through adenovirus-mediated expression elevated the water permeability of cellular membranes (*Jiang, 2009*). Conversely, down-regulation of *AQP1* has been associated with the protection of tumors against cytotoxic edema by maintaining extracellular acidification, thereby facilitating the metastasis of glioma (*Pedersen, Hoffmann & Mills, 2001*). However, Aishima et al. found that a significant association exists between low or negative expression of *AQP1* and certain unfavorable characteristics in intrahepatic cholangiocarcinoma, including larger tumor size and poorly differentiated histology. Additionally, a significant association was identified between decreased *AQP1* expression and the presence of lymph node metastasis. Interestingly, a reciprocal association was observed between *AQP1* expression and *MUC5AC*, a mucus core protein, and their distributions showed a complementary pattern. The downregulation of *AQP1* appeared to be closely linked to the expression of *MUC5AC* and the aggressive behavior of the tumor (*Aishima et al., 2007*). According to recent research, the expression of *AQP1* had been detected in biliary tract carcinoma (BTC), and its levels decreased as the carcinoma progresses in terms of invasion. These results suggested that *AQP1* expression held great

potential as a valuable prognostic marker and indicator of tumor invasion in BTC (*Sekine et al., 2016*). However, how *AQP1* affects WT migration is particularly important. EMT activation plays a crucial role in the metastasis of cancer cells, as it facilitates the transition of epithelial cells into a mesenchymal phenotype. Notably, attenuation of EMT triggers cytoskeletal rearrangement, further modulating cellular behavior. We examined the effect of *AQP1* overexpression on EMT in WT. We detected that the expression of biomarkers of EMT were reduced in WT. Current studies are focused on the process of cell migration, including the extension of plasma membrane protrusions or lamellipodia and filopodia at the "leading edge", the formation of new adhesion points at the front end of the cell, the contraction of the cell body, and the deadhesion of the cell tail. Among these cell migration processes, plasma membrane protrusion at the "leading edge" or the extension of lamellipodia and filopodia are the main driving forces (*Webb, Parsons & Horwitz, 2002*). The cytoskeleton plays a vital role in cell migration (*Seetharaman & Etienne-Manneville, 2020*). To visualize microfilament structures, Phalloidin-iFlour 488 was employed in the labeling of cells in each experimental group. Notably, the *AQP1* overexpression group exhibited significant changes in cytoskeletal microfilament organization. This suggests that overexpression of *AQP1* inhibited the migration of WT cells, possibly by affecting the cell microfilament structure and interfering with lamellipodia and filopodia formation. This study revealed that overexpression of *AQP1* inhibited WT cell migration and invasion, which might be related to interference with the formation of the cytoskeleton.

New research findings have illuminated the dynamic process of kidney development, specifically highlighting the transformation of cells from the cap mesenchyme, a unique progenitor state exclusive to the kidney, into an epithelial phenotype termed mesenchymal-to-epithelial transition (MET). Subsequently, these cells undergo differentiation into diverse epithelial cell types, playing a pivotal role in the development of tubular structures within the nephron. Disruptions in the critical transition process have been implicated in the pathogenesis of WT. In this particular study, early, middle, and late embryonic kidney samples, as well as WT samples, were subjected to sequencing analysis. The findings exhibited a sequential decline in mesenchymal-associated gene expression and a simultaneous increase in epithelial-associated gene expression during fetal kidney development. In contrast, the WT samples exhibited elevated expression levels of mesenchymal-associated genes and relatively reduced expression levels of epithelial-associated genes (*Wineberg et al., 2022*). We postulate that the low expression levels of *AQP1* in WT may inhibited the process of MET, while the overexpression of *AQP1* may potentially reduce the formation of mesenchymal in cells.

In conclusion, We identified co-expressed DEGs with three datasets GSE57269, GSE73209 and GSE66405 for WT, and predicted PPI networks, GO terms, KEGG pathways for co-expressed DEGs. The top five key genes were identified: *REN, NPHS2, SLC12A3, SLC12A1,* and *AQP1*. The overexpression of *AQP1* might inhibit WT cell migration and invasion, which could result from decreased EMT and dysregulated cytoskeleton formation. These findings offer valuable insights into the molecular mechanisms contributing to the onset and advancement of WT.

### Funding

This work was supported by the Key R&D Plan fund from Shandong Province (No. 2015GSF118130). The funders had no role in study design, data collection and analysis, decision to publish, or preparation of the manuscript.

### Grant Disclosures

The following grant information was disclosed by the authors:
Key R&D Plan fund from Shandong Province: 2015GSF118130.

### Competing Interests

The authors declare there are no competing interests.

### Author Contributions

- Hong Liu conceived and designed the experiments, performed the experiments, analyzed the data, prepared figures and/or tables, authored or reviewed drafts of the article, and approved the final draft.
- Chen Jin performed the experiments, prepared figures and/or tables, authored or reviewed drafts of the article, and approved the final draft.
- Xia Yang performed the experiments, prepared figures and/or tables, authored or reviewed drafts of the article, and approved the final draft.
- Nan Xia performed the experiments, prepared figures and/or tables, authored or reviewed drafts of the article, and approved the final draft.
- Chunzhi Guo analyzed the data, prepared figures and/or tables, authored or reviewed drafts of the article, and approved the final draft.
- Qian Dong conceived and designed the experiments, prepared figures and/or tables, authored or reviewed drafts of the article, and approved the final draft.

### Human Ethics

The following information was supplied relating to ethical approvals (i.e., approving body and any reference numbers):

Affiliated Hospital of Qingdao University granted Ethical approval to carry out the study within its facilities(Ethical Application Ref: QYFY WZLL 27451).

### Data Availability

The raw measurements are available in the Supplementary File.

### Supplemental Information

Supplemental information for this article can be found online at http://dx.doi.org/10.7717/peerj.16025#supplemental-information.

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
