# Peer review of "Identification of key genes and validation of key gene aquaporin 1 on Wilms’ tumor metastasis"

_PeerJ, doi:10.7717/peerj.16025_

## Round 0.1 · original submission · Major Revisions

· Academic Editor

Major Revisions

I am sending it back for revisions. Please revise and submit a rebuttal.

Reviewer 1 ·

Basic reporting

1.clear and professional English used: yes

2. literature reference are up do date : yes

3. article structure, figures , tables , raw data shared : yes

4. relevant to hypothesis

Experimental design

1. Original primary research within Aims and Scope of the journal.: yes


2. Research question well defined, relevant & meaningful. It is stated how research fills an identified knowledge gap: yes at very good extent.

3.Rigorous investigation performed to a high technical & ethical standard.: yes

4.Methods described with sufficient detail & information to replicate.: yes

Validity of the findings

yes , its novel finding and has potential use in therapeutics of Wilms tumor.


Conclusions are well stated, linked to original research question & limited to supporting results.: yes

Additional comments

its novel good article with novel findings. valid research findings.

Reviewer 2 ·

Basic reporting

This study aimed to identify key genes and molecular mechanisms associated with Wilms' tumor (WT) through bioinformatics analysis of gene expression data from the Gene Expression Omnibus (GEO) database. The study identified AQP1 as the most enriched key gene in WT and found that AQP1 overexpression inhibited WT cell invasion and migration by affecting epithelial-mesenchymal transformation (EMT) and cytoskeletal rearrangement. The study used various methods, including gene functional enrichment analysis, protein-protein interaction network construction, and wound healing and Transwell assays. The results suggest that AQP1 may be a potential therapeutic target for WT.

I have reviewed your manuscript and would like to provide feedback on how to improve it. Firstly, your study design approach is commendable, as it covers both in silico and in vitro experiments. However, there seems to be a discrepancy between your stated approach and your actual methodology. Based on my review of your raw data, it appears that you used a bottom-up approach to select genes, but presented it as a top-down approach in the manuscript. Your abstract, in line 29-30, states that "This study aimed to explore the key genes and potential molecular mechanisms through 30 bioinformatics and to verify the effects of AQP1 on WT metastasis." This perspective is repeated throughout the manuscript. However, after reviewing your DEGs table, I found many other genes that were more significant and had higher logFC values. Additionally, your network analysis highlighted more top genes for in vitro evaluation. Therefore, I suggest that you clarify this discrepancy in your methodology and results sections.

It is worth noting that a bottom-up approach is acceptable and can be useful for tracking gene behavior. However, when presenting the results in an article, it is essential to present them as they are.

Your figures and tables are well-matched with your text and provide sufficient clarity. However, I highly recommend adding a flowchart for your method strategy, which can help readers better understand your approach.

Furthermore, your literature references, especially in the discussion section, are not sufficient and outdated. Many recent studies have used similar concepts and methodologies in Wilms' tumor but yielded different results. Therefore, your discussion should be updated to include the involved genes and pathways and their differences. Your study highlights the importance of cell-cell communication, but previous studies have found additional pathways that may be relevant.

Overall, your manuscript has potential, but some improvements are necessary to make it more clear, comprehensive, and up-to-date.

Experimental design

-As two datasets, GSE57269 and GSE73209, were used in the study, it should be noted that each sample has a different source; one is from a xenograft, and the other is from fresh frozen tissue. It is important to consider that various factors such as race, age, and gender can influence gene expression when evaluating gene expression. Using different sources of data may introduce biases in the results. Additionally, the number of normal samples in the GSE73209 dataset is not consistent with the others. Therefore, it is recommended to consider changing the datasets to ones that are more comparable. As a suggestion, the GSE66405 dataset could be considered.

-The clarity of the text from line 239 to 244 is insufficient.

- In line 94 of the manuscript, it says that two sets of gene expression data were downloaded from a database called GEO. However, later in the manuscript, it is mentioned that GEO2R was used to analyze the data. GEO2R is an online tool that can directly analyze the data in GEO without needing to download it first. The tool can generate the list of differentially expressed genes and graphs, which can be downloaded online. Therefore, it is not clear why the data was downloaded separately before using GEO2R.

Validity of the findings

- The DEGs list indicates that AQP1 has four different isoforms, each with varying expression levels. The isoform that meets the selection criteria was included in the analysis. To ensure clarity in the manuscript, it would be beneficial to refer to the specific gene ID for AQP1.

Additional comments

- please provide all your supplementary file names in english.

Reviewer 3 ·

Basic reporting

Please see the Additional comments.

Experimental design

Please see the Additional comments.

Validity of the findings

Please see the Additional comments.

Additional comments

The authors combined bioinformatics and cell experiments to explore the role of AQP1 in WT pathogenesis. Overall, this study is suitable for publication, only if the authors address the following issues:

1. Throughout the manuscript, it seems better to use Grammarly (https://www.grammarly.com/) to check & correct potential grammatical errors. For example,
1.1 In INTRODUCTION, it seems better to change "Although much progress has been made in WT treatment in recent decades, the survival rate for patients with WT still needs to improve 2.20% of patients relapse after first-line therapy and up to 25% of survivors report severe late morbidity of treatment" into "Although much progress has been made in WT treatment in recent decades, the survival rate for patients with WT still needs to be improved: 2.2% of patients relapse after first-line therapy and up to 25% of survivors report severe late morbidity of treatment", which would be clearer. Similarly, it seems better to change "Novel biomarkers, therefore, are urgently needed in order to diagnose the disease early and develop novel target-specific therapies". Likewise, it seems better to change "It is very important to improving prognosis for WT patients by improving treatment based on clinical and biological risk factors, and further stratification of current treatment options based on tumor biology" into "It is very important to improve the prognosis for WT patients by promoting the treatment based on clinical and biological risk factors, and further stratification of current treatment options based on tumor biology".
1.2 In RESULTS, it seems better to change "The functional annotations of top 10 key genes were performed by DAVID, GO analysis results revealed that AQP1 was the most involved key gene among the most significantly enriched functions" into "The functional annotations of top 10 key genes were performed by DAVID, whose GO analysis results revealed that AQP1 was the most involved key gene among the most significantly enriched functions." Similarly, it seems better to change "..., indicative of AQP1 overexpression significantly inhibited the migration ability of WT cells" into "..., indicating that AQP1 overexpression significantly inhibited the migration ability of WT cells"; it seems better to change "..., suggestive of AQP1 overexpression significantly inhibited the invasion capacity of WT cells" into "..., suggesting that AQP1 overexpression significantly inhibited the invasion capacity of WT cells".
1.3 In DISCUSSION, it seems better to change "In the GO analysis, it was found that AQP1 was mainly involved in the he transmembrane transporter activity" into "In the GO analysis, it was found that AQP1 was mainly involved in the transmembrane transporter activity".

2. In all FIGURES, it would be more clear and readable to BOTH use high-resolution SVG/PDF format (instead of low-resolution PNG) AND expand on figure legends by explaining the meanings of colors, groups, lines, and abbreviations. For example,
2.1 In the legend of Figure 1A–B, please mention the meaning of green & red dots and X & Y axes; in the legend of Figure 1C, please mention the meaning of 552, 86, and 182. (please see how all elements in a volcano plot were explained by this article: PMID_27518660)
2.2 In Figure 2A, please change the last four bluish colors, which seem difficult to distinguish from each other; in Figure 2B, please delete unnecessary information (such as "hsa00330", "3.488372093", "0.026446644", etc.); in the legend of Figure 2B, please mention the meaning of BOTH different sizes & colors of the nodes AND the X axis.
3.3 In the legend of Figure 3, please mention the meaning of the different colors of the nodes (in Figures 3E–F) and the distinct types & colors of the edges or lines (in Figures 3A & 3F).

These revisions would greatly help readers, who do not specialize in bioinformatics, to understand the results and their implications easily and efficiently.

3. In ABSTRACT:
3.1 In Background, it seems better to replace the two sentences with one sentence talking about why researching Wilms' tumor is important AND the other sentence pointing out what is the research gap — a gap that has not been filled by previous studies but is being filled by the current research (FYI: a pattern like PMID_34715879, PMID_34384362, or PMID_35965679).
3.2 In Methods, it seems more rigorous to mention the two GSE57269 and GSE73209. In addition, it seems better to change "Phalloidin-iFlour 488 was used to stain the cytoskeleton to observe the effect of the cytoskeletal microfilament structure" into "Phalloidin-iFlour 488 was used to stain the cytoskeleton to observe how AQP1 overexpression affects cytoskeletal microfilament structure", which would be clearer.
3.3 In Results, it seems better to change "AQP1, as the most enrichment key gene, were identified in WT" into "AQP1 was one of the most differentially expressed genes between normal and WT tissues, based on two GEO datasets", which would be more accurate.
3.4 In Results, it seems better to change "AQP1 was expressed at significantly lower levels in WT than in normal tissue" into "AQP1 was validated to be expressed at significantly lower levels in WT than in normal tissue", which would be more cohesive.
3.5 In Results, it seems better to change "The scratch healing rate and the number of perforated cells were significantly lower in the AQP1 overexpression group than in the negative control group. We detected that the expression of biomarkers of epithelial-mesenchymal transformation (EMT) were reduced in the AQP1 overexpression group. Conspicuous lamellipodial protrusions formation was reduced in the AQP1 overexpression group, suggestive of diminished WT cell invasion and migration" into "Compared with the negative control, AQP1 overexpression significantly lowered the scratch healing rate of cells and the number of perforated cells. Consistently, the overexpression reduced the expression of biomarkers of epithelial-mesenchymal transformation (EMT). Also, AQP1 overexpression decreased the formation of conspicuous lamellipodial protrusions formation, suggesting diminished WT cell invasion and migration", which would be more coherent and cohesive.
3.6 In Conclusion, it seems better to change "AQP1 overexpression inhibited WT cell invasion and migration by affecting EMT and cytoskeletal rearrangement" into "AQP1 overexpression inhibited invasion, migration, EMT, and cytoskeletal rearrangement of WT cells, indicating that AQP1 plays a role in the pathogenesis of WT", which would be more neutral and not exaggerated.

4. In INTRODUCTION:
4.1 In Paragraph 1, it seems clearer to rewrite "But the introduction of biology-driven approaches to risk stratification has been slower in WT than in other childhood tumors" by clarifying "biology-driven approaches".
4.2 In Paragraph 2, it seems clearer to rewrite "With the development of modern genetic technology, BCOR, MAP3K4, BRD7, CREBBP and HDAC4 have been found in WT that are involved in histone modifications during kidney formation" by clarifying whether "BCOR, MAP3K4, BRD7, CREBBP and HDAC4" OR "WT" "are involved in histone modifications during kidney formation". If "BCOR, MAP3K4, BRD7, CREBBP and HDAC4" "are involved in histone modifications during kidney formation", how is the expression of these genes linked to WT?
4.3 In Paragraph 3, it seems more logical to add the background of AQP1 and to highlight the research gap in the linkage between AQP1 and WT.
4.4 In Paragraph 4, it seems more logical to replace this section with a part introducing how WT and/or AQP1 are connected to metastasis & EMT — the major phenotypes of this study.

5. In METHODS, please justify why "A high confidence score of 0.7 was set as the minimum" and why "Molecular Complex Detection (MCODE), a Cytoscape plug-in, was used to identify more significant modules of the PPI network. The criteria were set as follows: degree cutoff=2, node score cutoff=0.2, K-core=2, and max depth=100".

6. In RESULTS:
6.1 It would be clearer to end each paragraph in RESULTS with one sentence: "Together, these results suggest that ..." (a pattern like PMID: 34715879, PMID: 34384362, PMID: 35965679, and PMID: 34537192), summarizing a paragraph AND highlighting the implications of all results in the paragraph.
6.2 In "Identification of WT-related genes", at the end of "A total of 1195 DEGs in WT were screened, including 40 upregulated genes and 1155 downregulated genes", it would be more rigorous to mention the threshold of the p & fold change values.
6.3 In "Key genes were selected by the PPI network", it seems better to change "The STRING database was applied to construct a PPI network of these 86 DEGs to explore the core modules and key genes that played the most importance in modular genes" into "The STRING database was applied to construct a PPI network of the 86 coexpressed DEGs to explore core modules and key genes, both of which could play a central role in the network", which would be clearer. Similarly, it seems better to change "The PPI network of 86 DEGs for GSE57269 and GSE73209, including 85 nodes (genes) and 119 edges (interactions), was screened by STRING and Cytoscape" into "The PPI network, including 85 nodes (genes) and 119 edges (interactions), was further analyzed by two built-in algorithms within Cytoscape", which seems clearer and more cohesive.
6.4 In "Key genes were selected by the PPI network", please mention why the authors "retrieved" the modules; in other words, please highlight that AQP1 is a gene in the modules which could play an important role in the network.
6.5 In "X2K construction and visualization of the upstream regulatory network", please explain why the authors conducted the X2K analysis, whose results did not seem related to AQP1 or WT.
6.6 To highlight what the authors wrote in "GO enrichment analysis of the key genes", it seems better to use color to highlight the "AQP1"s in Table 1.
6.7 In "GO enrichment analysis of the key genes", it seems better to change "... AQP1 was the most involved key gene among the most significantly enriched functions. AQP1 had the highest participation in BP, CC and MF of top 5 with p value" into "... AQP1 could be a key gene. Because AQP1 was predicted to be the most significantly associated with the top five enriched terms in BP, CC, and MF", which seems more cohesive.
6.8 In "Expression of the key gene AQP1 in WT", it seems better to change "In general, AQP1 expression levels are significantly downregulated in WT cells. To investigate the expression levels of AQP1 in WT Wit-49 cells and HK12 normal kidney cells" into "In the GEO dataset analysis, AQP1 was significantly downregulated in WT tissues. To validate this finding, the expression levels of AQP1 in WT Wit-49 cells and HK12 normal kidney cells", which would be clearer and more cohesive. In addition, please delete "The expression levels of AQP1 were significantly downregulated at both the mRNA and protein levels in the WT Wit-49 cells compared with normal kidney HK12 cells", which seems a repetition of the sentences after it.
6.9 In "Expression of the key gene AQP1 in WT", it seems better to change "To further verify the role of AQP1 in renal tubules and WT cancerous tissues" into "To further explore the expression patterns of AQP1 in renal tubules and WT cancerous tissues". Otherwise, please mention how the IHC staining results could "verify the role of AQP1".
6.10 In "Overexpression of AQP1 suppresses the migration and invasion of WT cells in vitro", it seems better to EITHER delete "Cytoskeletal assays of AQP1-overexpressing cells and negative control cells were visualized using the LSCM. Phalloidin-iFlour 488 staining of F-actin revealed the cytoskeleton, which appeared as green fluorescence, and DAPI staining was used to stain the nuclei. Then, we overlapped the two parts" OR put it in METHODS. Instead, it would be more logical to explain why the authors did cytoskeletal assays (perhaps because the assays suggest cell migration potential?)

7. In DISCUSSION:
7.1 It seems better to integrate the first and second paragraphs into one.
7.2 In Paragraph 2, it seems better to delete "We also tried to investigate how AQP1 involved in the effects of cell migration and invasion ability", which could be a repetition of the sentences before it.
7.3 In the final paragraph, it seems more accurate to rewrite "In conclusion, we identified DEGs, PPI networks, GO terms, KEGG pathways" by expanding on whose "DEGs, PPI networks, GO terms, KEGG pathways" were identified. In addition, it seems better to change "The overexpression of AQP1 might inhibit WT cell migration and invasion, which is related to decreased EMT, thereby interfering with the formation of the cytoskeleton" into "The overexpression of AQP1 might inhibit WT cell migration and invasion, which could result from decreased EMT and dysregulated cytoskeleton formation", which would be clearer and more accurate.

---

## Round 0.2 · Minor Revisions

· Academic Editor

Minor Revisions

Reviewers mentioned this, but editing is still needed for this sentence: "To further explore the expression patterns of AQP1 in renal tubules and WT cancerous tissues."

I also suggest that you consider this substitution in your abstract:
Use these sentences "AQP1 overexpression
Significantly reduced the migratory and invasive capacity of Wit-49 cells, as evidenced by reducing the scratch healing rate and the number of perforated control cells by Wit-49 cells. AQP1 overexpression also reduced the expression of biomarkers of epithelial-mesenchymal transformation, decreased levels of vimentin and N-cadherin and increased expression of E-cadherin, resulting in decreased formation of conspicuous lamellipodial protrusions, characteristic of diminished WT cell invasion and migration."
to replace
"Compared with the negative control, AQP1 overexpression significantly lowered the scratch healing rate of cells and the number of perforated cells. Consistently, the overexpression reduced the expression of biomarkers of epithelial-mesenchymal transformation (EMT). Also, AQP1 overexpression decreased the formation of conspicuous lamellipodial protrusions formation, suggesting diminished WT cell invasion and migration."

Also, are there any drugs that could be candidates to increase AQP1 expression as a treatment for WT?"

Reviewer 1 ·

Basic reporting

Article is up to standerd

Experimental design

Experimental design is good. Within scope of journal

Validity of the findings

Findings are valid, it has further scope of research. Especially in era of targeted therapy

Additional comments

Nil

Reviewer 3 ·

Basic reporting

Thank the authors for their efforts to reply to all of my comments. The current version would be suitable for publication in this journal.

Experimental design

N/A

Validity of the findings

N/A

Additional comments

N/A

---

## Round 0.3 · accepted · Accept

· Academic Editor

Accept

All revisions are done as per the reviewers and editorial board members and this paper is accepted as is.